# [Citation needed] Data usage and citation practices in medical imaging conferences

**Théo Sourget**[1,2]                        TSOU@ITU.DK
**Ahmet Akkoç**[1,3]
**Stinna Winther**[1]
**Christine Lyngbye Galsgaard**[1]
**Amelia Jiménez-Sánchez**[1]
**Dovile Juodelyte**[1]
**Caroline Petitjean**[2]
**Veronika Cheplygina**[1]                     VECH@ITU.DK
[1] *IT University of Copenhagen, Denmark*
[2] *University of Rouen, France*
[3] *ZiteLab ApS, Denmark*

**Editors:** Accepted for publication at MIDL 2024

## Abstract

Medical imaging papers often focus on methodology, but the quality of the algorithms and the validity of the conclusions are highly dependent on the datasets used. As creating datasets requires a lot of effort, researchers often use publicly available datasets, there is however no adopted standard for citing the datasets used in scientific papers, leading to difficulty in tracking dataset usage. In this work, we present two open-source tools we created that could help with the detection of dataset usage, a pipeline[1] using OpenAlex and full-text analysis, and a PDF annotation software[2] used in our study to manually label the presence of datasets. We applied both tools on a study of the usage of 20 publicly available medical datasets in papers from MICCAI and MIDL. We compute the proportion and the evolution between 2013 and 2023 of 3 types of presence in a paper: cited, mentioned in the full text, cited and mentioned. Our findings demonstrate the concentration of the usage of a limited set of datasets. We also highlight different citing practices, making the automation of tracking difficult.

**Keywords:** Bibliometrics, citations, datasets, medical imaging, data re-use, annotation

## 1. Introduction

While the increased usage of open data is a positive development, we hypothesize it might introduce a shift in the targeted applications. For example, (Varoquaux and Cheplygina, 2022) show that since the Kaggle lung cancer challenge in early 2017 (Buckeye et al., 2017), there has been a disproportionate increase in machine learning papers on lung cancer, while many of the proposed solutions do not differ in practice. A similar concentration on fewer datasets has also been found in machine learning (Koch et al., 2021). Another medical

---

1. https://github.com/TheoSourget/Public_Medical_Datasets_References
2. https://github.com/TheoSourget/pdf_annotator

imaging example is the segmentation of cardiac ventricles, addressed with multiple competitions (Bernard et al., 2018b; Suinesiaputra et al., 2012; Petitjean et al., 2015; Campello et al., 2021). The latest competition achieved highly accurate results and commercially available software exists (Wu et al., 2024), yet the application still remains popular for evaluating novel algorithms. Moreover, while the availability of a public dataset is a positive step towards getting a problem addressed by the community, the choice of a single dataset for evaluation also results in an overestimation of performances leading to a gap when applied on a different one (Wu et al., 2021).

There is a need to analyse research within a field to understand the progress being made, but next to surveys focused either on methods (Litjens et al., 2017; Cheplygina et al., 2019; Budd et al., 2021) or on datasets within a specific application (Daneshjou et al., 2021; Wen et al., 2022), we find few studies on understanding *dataset use* beyond their initial release in the field. We believe this is in part due to identifying dataset usage, as datasets may be used without corresponding citations, and vice versa. Our contributions, aiming to achieve this identification of dataset usage are as follows: **(1)** We present a fully automated pipeline for quantifying dataset usage based on the analysis of references and the paper full text. **(2)** We present an open-source annotation tool which allows for validation of the method, and can aid in tracking dataset usage in research papers. **(3)** We apply both tools to study the usage of several popular segmentation and classification datasets and their usage in MICCAI and MIDL conference papers between 2013 and 2023. **(4)** We discuss the limitations of our study and tools, display additional practices we found during our study, and provide recommendations to ease the tracking of datasets.

## 2. Related Work

Meta-research papers in medical imaging often focus on **methods**, for example surveys on deep learning (Litjens et al., 2017), different types of supervision (Cheplygina et al., 2019), human-in-the-loop methods (Budd et al., 2021) and so forth. As a by-product of annotating and categorizing papers, some surveys also provide lists of commonly used datasets (Çallı et al., 2021b).

More recently some **dataset**-focused reviews started to emerge, in particular for dermatology (Daneshjou et al., 2021; Wen et al., 2022) and ophtalmology (Khan et al., 2021). These reviews focus on the type of data that is available, and find various biases in the patient populations, and/or that metadata about the patient demographics is missing. However these papers do not examine dataset use.

Perhaps at the intersection of datasets and methods, there is work focusing on challenges (Eisenmann et al., 2022) which review participation in medical image competitions at MICCAI and ISBI. Such competitions are often seen as one of the drivers of publicly available datasets, but the impact of these datasets beyond these competitions is not known.

The closest to our work are studies that examine dataset usage in other published works. (Koch et al., 2021) analyse dataset usage on PapersWithCode across various applications of machine learning, and find that the diversity of datasets used is decreasing. Within medical imaging, Heller et al (Heller et al., 2019) examined the role of publicly available data in MICCAI papers between 2014 and 2018, and found among others that over 20% of papers using public data did not cite the dataset. Simkó et al (Simko et al., 2022) examined

reproducibility in MIDL papers between 2018 and 2022 and found that papers using public datasets are becoming more common but without proper citations or links.

## 3. Quantifying medical image dataset use

We propose tools to evaluate the presence of datasets using the following definitions: a dataset is **cited** if its paper is present in the reference section, **mentioned** if its name, aliases or URL are in the abstract or in a section of the paper associated with the method or results (i.e., not in a related work or discussion sections), in a table or figure captions or in a footnote, showing an actual **usage**. We show our pipeline in Figure 1. There are four main components: user input about the list of venues and datasets to track, the open citation index tool OpenAlex (Priem et al., 2022), the full texts of the papers and GROBID, a tool to extract information from scholarly documents. We used OpenAlex because it has an official freely accessible API aggregating and standardizing information from multiple sources such as arXiv, Crossref and Pubmed, and we aimed to create a generalizable process that could be complemented with other tools. We compared it with OpenCitations to evaluate their completeness using the citations returned for a set of cardiac segmentation datasets. We found that 97% of the citations returned by OpenCitations were also returned by OpenAlex while only 84% returned by OpenAlex were returned by OpenCitations; thus we chose OpenAlex for our pipeline.

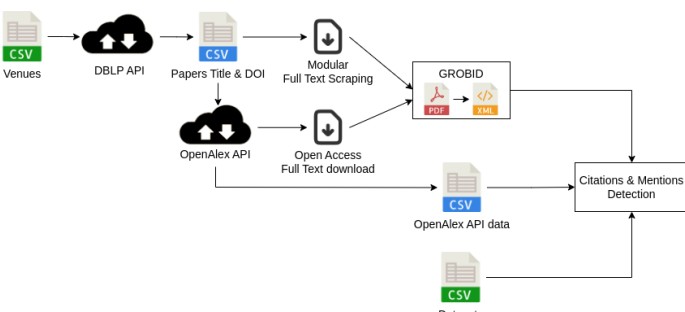

Figure 1: Pipeline to detect dataset presence and usage. Green CSV represents user input, blue CSV represents extracted data

First, we ask the user to specify the list of venues and datasets to track. This includes a dataset name, any aliases referring to the same datasets, and the titles and DOIs of papers associated with these datasets.

We use the venue list to fetch the list of papers from these venues using the DBLP API and store the papers' titles and DOIs. We then use the paper DOI (or title if the DOI is not available) to query the OpenAlex API to get the following: (i) list of referenced papers, (ii) list of words in the abstract, and (iii) open access link to the paper's full text, an example of data from OpenAlex is shown in Appendix D. We then try to fetch the paper's full text. If this step fails, we complement this step with a custom scraping tool. This step can be easily replaced for different venues, as long as the paper PDFs are stored in the same

folder, In this step some data cleanup may be necessary to remove duplicates created from the combination of PDF extracted with OpenAlex and with the scraping tool. We then convert the PDF to an XML file using GROBID, a library applying a cascade of machine learning models that first segment the document in different structures like header, full text or reference and then specific models tuned for each type of structure to extract content. This allows to detect different paper sections, while keeping information about figures, tables and footnotes, which were lacking in alternative tools such as PyPDF, an example of the data obtained with GROBID is shown in Appendix E.

We use the dataset list to detect their citations and mentions. We detect citations in two ways: based on the dataset's DOI converted to an OpenAlex ID, and based on matching the dataset paper titles to the references sections of the full text. We detect dataset mentions by searching for the dataset's name, aliases or URL in certain fields of the full text. Examples of different types of citations and mentions can be seen in Appendix C.

Finally, we assign each dataset presence to one of the following types: only cited, only mentioned, and both cited and mentioned.

## 4. Annotation tool for paper PDFs

We also present our PDF annotation tool made with Streamlit, a Python library to easily create web apps. We used it to verify our detection process and therefore we designed it to fulfil two needs: having multiple users annotate the same project easily and being able to handle a large number of PDF files. While it was used to annotate datasets' presence in scientific papers, it can be extended to any PDF annotation task.

Once the software is installed on a local server, a user can create an annotation project by uploading the PDFs and choosing up to two initial sets of labels. In our study, the first set of labels corresponds to the list of datasets to detect and the second set is the list of locations a mention could be classified into (E.g. Abstract, Introduction, Method). While the second set of labels is fixed, the first one is not and new values can be added at any point during the labelling.

We also wanted to ease the annotation by multiple users. At the creation of the project, the owner can upload a file containing the division of the papers into different groups. This way, users can find the papers they were assigned to by selecting the right group on the annotation page. Finally, when the annotations are downloaded from the server, a file per person is obtained allowing more data processing afterwards.

## 5. A case study on publicly available medical datasets

### 5.1. Data selection

We apply our tools to a set of 20 publicly available medical datasets for both classification and segmentation of various organs shown in Appendix A. We initially tried a systematic procedure of identifying datasets via Google datasets and OpenAlex. However, this resulted in many poorly documented datasets (particularly on COVID-19) which did not have distinct names, and of which we could not trace whether they were in part duplicated from other datasets. Therefore, we selected datasets based on a combination of prior knowledge of the authors and consulting recent surveys in medical imaging which provided a table or list of

datasets (Hesamian et al., 2019; El Jurdi et al., 2021; Niyas et al., 2022; Qureshi et al., 2022; Çallı et al., 2021a; Guan and Liu, 2021). In order to obtain enough data to analyse, we took the following aspects into account for the selection: presence of a paper linked to the dataset available in OpenAlex, year of publication, having some citations in OpenAlex, and having a unique name or acronym to make the detection process more reliable. We chose to analyse papers from two major conferences about medical image analysis, MICCAI and MIDL, so that the papers are more likely to contain the presence of such datasets.

We identified 4835 papers in total (4569 from MICCAI and 266 from MIDL), however, 44 were discarded as we could not obtain information on the content of the paper or the list of references. We categorize the remaining papers in three groups, where for each group we slightly adjusted our processing due to the missing data:

- Group 1 with n=2327 papers either have all all information, or only miss the abstract from OpenAlex. In this case, we analyze the abstract from the full text of the paper.

- Group 2 with n=2237 where full text is not available, but we can still detect dataset mentions using the OpenAlex abstract. This is an important limitation, as the abstract does not always mention the datasets used. All the papers in this group are from MICCAI, showing the usefulness of the modular part to obtain the full texts. Unlike MICCAI papers, the structure of the PMLR website and the complete open access of PDFs made possible the development of the scraping tool for all MIDL full texts while they were not accessible from OpenAlex.

- Group 3 with n=227 papers which do not have references in OpenAlex. We therefore detect citations only with our simpler matching of dataset papers' titles with Grobid, which may result in less accurate detection. A majority of papers from this group are from MIDL as the information for papers from this conference is almost absent from OpenAlex. This shows that the download and analysis of the full text is a crucial and needed aspect of our method.

### 5.2. Concentration of research on few datasets

Although we considered the number of citations in OpenAlex to make the first selection of datasets, some datasets had very low numbers of citations and mentions in MICCAI and MIDL. We only present in Figures 2 and 3 results for eight datasets with the highest usage or that exemplify one of our conclusions, a more complete version including all the datasets can be found in Appendix B. This result may highlight the focus on some particular datasets also shown in (Koch et al., 2021) when using publicly available data, especially for datasets for the same task (cardiac segmentation and chest classification) as ACDC and M&Ms. This is also visible in Figure 2 with the large gap between the count of citations and mentions for BRATS and the rest of the most present datasets.

Note the difference in growth between the datasets, which might suggest a snowball effect where popular datasets become even more popular. This seems to be the case for BRATS, ACDC or Chexpert which have a very strong growth in citations and mentions. For other datasets like LIDC-IDRI or DRIVE, the number of citations and mentions is more gradual and even stagnates for DRIVE. Multiple factors can impact the popularity of a dataset, one of the most straightforward is the year of publication but while Chexpert and

PadChest have been released at the same time, the second is almost absent from our list of papers. Therefore, other aspects such as how the dataset is updated or has a competition been organized with the dataset could be an explanation for such differences.

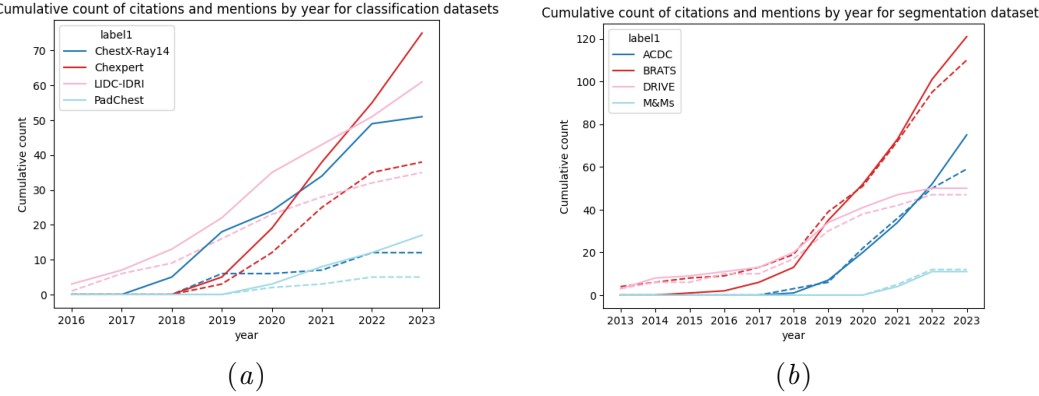

Figure 2: Cumulative counts per year of dataset citations (full line) and mentions (dashed line) for classification datasets (a) and segmentation datasets (b).

### 5.3. Lack of citation standards leads to difficulty in tracking usage

A dataset's citation doesn't necessarily imply actual usage and not all used datasets are cited in the references section. We further analyze this difference between mention and citations with Figure 3 in which we assign each presence of a dataset in a paper to one of the categories described in Section 3. We found out that even if there is variability in the groups' proportion for each subset, we can observe that almost every subset has more than 25% of datasets being only cited and around 10% being only mentioned. We considered papers from the "Only Cited" group as not using the dataset while citing it in the introduction or related works, mostly for general statements about machine learning usage in the medical sector. However, 132 papers out of 233 miss the full text and therefore only the abstract is used to detect the mention, a fraction of these papers could therefore mention the dataset and use it but the lack of information prevents our tool from detecting it. On the other hand, the "Only Mentioned" group mostly represents papers that are using a dataset without citing the associated paper. These two groups display the lack of standards to indicate the usage of a dataset such that it can be easily tracked. It also supports our approach to analyze part of the full text to determine such a usage.

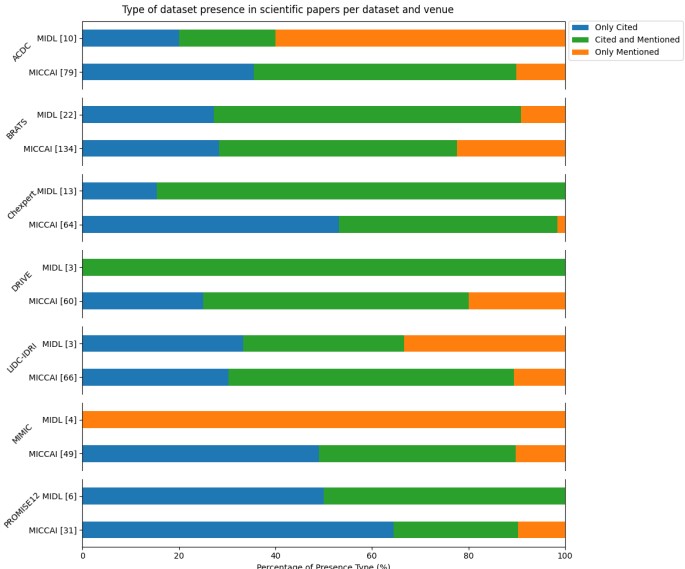

Figure 3: Type of presence per dataset and venue. The number in [] indicates the total number of papers for this subset. The "Only Cited" group in blue represents papers that cite a dataset without having a mention detected and therefore may not use it. The "Only Mentioned" group in orange represent the bad citation practice as the usage would not be detected by tools tracking the citations. The "Cited and Mentioned" group in green represent the best practice.

## 6. Discussion

We presented two open-source tools for the detection of dataset usage in scientific papers and applied them to a case study on publicly available medical datasets. We show that papers in major medical conferences tend to use a limited set of datasets, especially for papers addressing the same task. We also found that the lack of citation standards for dataset usage makes tracking such usage difficult, in particular due to (i) papers citing a dataset's paper without mentioning it in particular sections, indicating a non-usage, and (ii) papers mentioning a dataset without citing its paper, which classical bibliometric tools like OpenAlex can not detect.

Our study is limited to a set of datasets and venues manually selected and may therefore be biased by this selection. We also did not try to disambiguate between different datasets versions (for example different years of BRATS or datasets with similar names) due to already having difficulties with identifying these more-easily-identifiable datasets, it could however be valuable information to not overestimate the usage of a dataset or distinguish various tasks present across different version of a dataset. Doing a study on more datasets, venues and tasks would strengthen the outcome of our work. While datasets can be cited but not an associated paper, OpenAlex only keeps track of citations to papers. It is an

important limitation and therefore a more precise matching of citations using GROBID could be a solution to track citations without a paper like it can be for Kaggle datasets.

Our method relies on regex matching and their location, it makes our tool usable to other data easier as only some information needs to be changed. We did not use text classification methods based on deep learning, such as fine-tuning a model pre-trained on scientific data like (Beltagy et al., 2019). While this could result in better performances, it implies a fine-tuning for every new set of datasets, reducing the applicability of our tools to new settings.

While doing this study we had some anecdotal findings that we do not report on in the paper, but which we feel may warrant further study.

- We saw the number of citations a paper has doubled, from 10 to 20, in 2019. This is likely because until 2019 MICCAI used to include citations in the 8-page limit. Relaxing such page restrictions may incentivize authors to not omit dataset citations.

- We found many instances of papers associated with Github repositories that were promising the code to be available upon acceptance, but never actually did this.

- We found cases where a "backup" of a dataset on Kaggle was cited as if it were the original source. The dataset was often stripped of its metadata and license information, and it was not clear whether the data was exactly the same or a derivative of the original, for a longer discussion please see (Jiménez-Sánchez et al., 2024).

- We discovered that ACDC is a popular name, as it can refer to the Automated Cardiac Diagnosis Challenge (Bernard et al., 2018a) but also to the Adverse Conditions Dataset with images of streets (Sakaridis et al., 2021) or to the Automatic Cancer Detection and Classification in Whole-slide Lung Histopathology challenge (Li et al., 2018).

We believe that better knowledge and therefore easier access to dataset usage information are needed. In addition to giving due credit to the creators of the dataset, it can raise awareness of the overuse of a particular dataset for a task, which could have a negative impact on real performance, but also an over-representation of a task in regards of real clinical needs. Working towards the adoption of a standard for indicating the usage of a dataset seems to be an essential step to achieve this objective. As examples, NeuroImage has a specific section on data availability at the end of each manuscript, and in 2023, MICCAI added the obligation to declare "the data origin, data license, and (when appropriate) ethics application number for any public or private data used in the preparation of the paper". While such requirements will not solve all the issues at hand, we believe that including a "Data availability" section could be an easy solution to put in place that would pave the way towards more systematic ways of determining the usage of a dataset. There are of course still many unanswered questions as to how exactly we want to implement this, for example what to do in cases of derivative datasets, synthetic data, and so forth, which we hope we can discuss together as a community.

## Acknowledgments

Danish Data Science Academy DDSA-V-2022-004 and DFF - Inge Lehmann 1134-00017B

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

## Appendix A. List of selected datasets

Table 1: Summary of selected datasets

| Dataset | Organ | Published | Modality |
| --- | --- | --- | --- |
| **Segmentation datasets** | | | |
| ACDC (Bernard et al., 2018a) | Cardiac | 2017 | MRI |
| M&Ms (Campello et al., 2021) | Cardiac | 2021 | MRI |
| RVSC (Petitjean et al., 2015) | Cardiac | 2015 | MRI |
| STACOM'11 (Suinesiaputra et al., 2012) | Cardiac | 2011 | MRI |
| Sunnybrook (Radau et al., 2009) | Cardiac | 2009 | MRI |
| BRATS (Menze et al., 2015) | Brain | 2014 | MR |
| DRIVE (Staal et al., 2004) | Eye | 2004 | Fundus |
| CBIS-DDSM (Lee et al., 2017) | Breast | 2017 | Mammography |
| PROMISE12 (Litjens et al., 2014) | Prostate | 2014 | MR |
| **Classification datasets** | | | |
| ChestX-Ray14 (Wang et al., 2017) | Chest | 2017 | X-rays |
| Chexpert (Irvin et al., 2019) | Chest | 2019 | X-rays |
| LIDC-IDRI (Armato et al., 2011) | Chest | 2011 | CT |
| MIMIC (Johnson et al., 2019) | Chest | 2002 | X-rays |
| PadChest (Bustos et al., 2020) | Chest | 2019 | X-rays |
| VinDr-CXR (Nguyen et al., 2022) | Chest | 2020 | X-rays |
| CADDementia (Bron et al., 2015) | Brain | 2015 | MRI |
| CAMELYON (Litjens et al., 2018) | Breast | 2018 | whole-slide images |
| MRNet (Bien et al., 2018) | Knee | 2018 | MRI |
| PAD-UFES-20 (Pacheco et al., 2020) | Skin | 2020 | Phone picture |
| PROSTATEx (Armato et al., 2018) | Prostate | 2018 | mpMRI |

## Appendix B. Figures with original set of datasets

The following figures are the same as for Figures 2 and 3 without removing the datasets we considered not having enough matching. The non-presence of a dataset in one of the figures means that no paper contained a matching for this dataset.

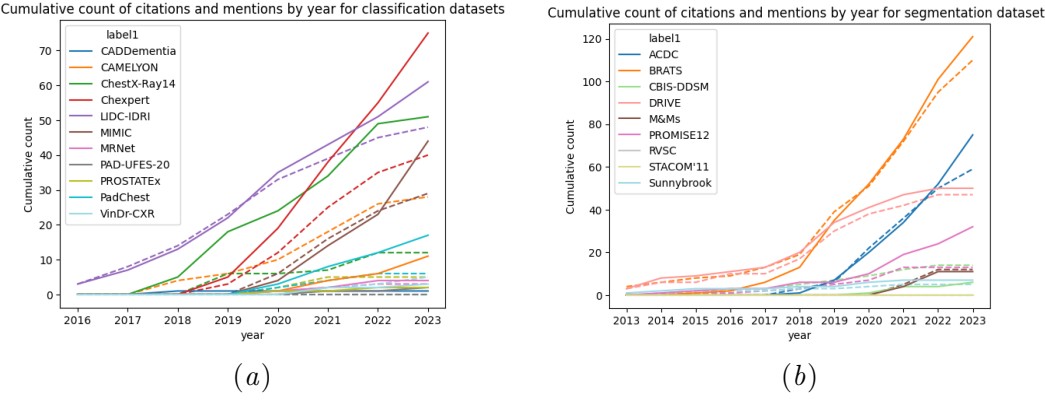

Figure 4: Cumulative counts per year of dataset citations (full line) and mentions (dashed line) for classification datasets (a) and segmentation datasets (b).

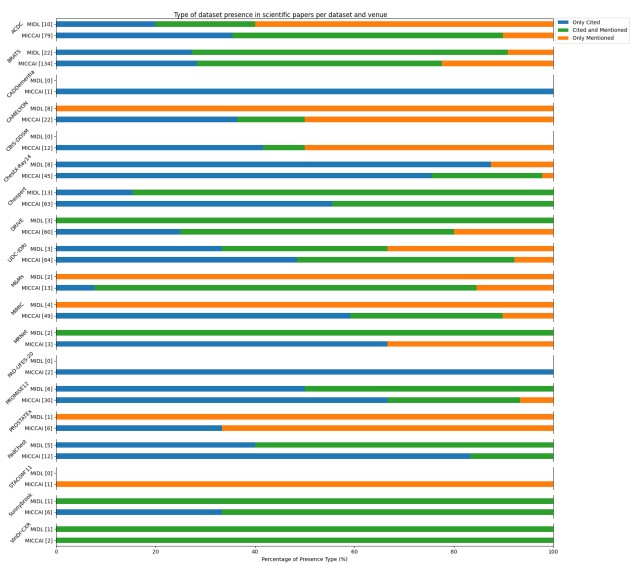

Figure 5: Type of presence per dataset and venue. The number in [] indicates the total number of papers for this subset.

## Appendix C. Example of dataset presence

### C.1. Citations

2015; Zhou et al., 2016). WSL has been applied in a wide range of medical-imaging applications, including the detection of lung disease in chest X-ray images (Wang et al., 2017; Yao et al., 2018; Tang et al., 2018; Ma et al., 2019; Liu et al., 2019; Guan et al., 2018), diagnosis of injuries from

Figure 6: Citation of a dataset without mention (Wang et al., 2017, ChestX-Ray8) in the background section for a demonstration of previous use

1. ISLES challenge (2017), http://www.isles-challenge.org/ISLES2017/
2. BRATS challenge (2018), https://www.med.upenn.edu/sbia/brats2018.html
3. ISLES challenge (2018), http://www.isles-challenge.org/ISLES2018/

Figure 7: Citation with a link to the datasets and not to the paper as indicated in BRATS guideline

### C.2. Mentions

**In text:**

**Diseased MRI Data.** For additional experiments, two brain disorder MRI datasets are used. First, we train the model with BRATS 2018 [2,11] dataset for brain tumor MRI generation, using 210 subjects in the training dataset labeled

Figure 8: Mention and citation to the papers of BRATS, following guidelines from the challenge

We also use the low grade glioma cases from the multimodal *Brain Tumor Segmentation* (BRATS) 2015 challenge[2]. This data contains 54 volumes imaged in

Figure 9: Mention of BRATS without a proper citation but only a footnote with a link to the data

**In figures and tables:**

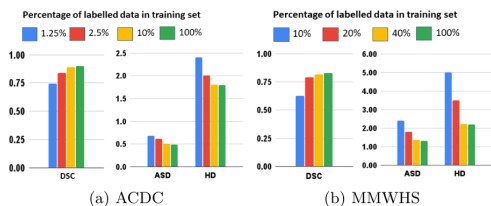

Fig. 2: Quantitative segmentation performance of our proposed method using different percentage of labelled data of ACDC and MMWHS.

Figure 10: ACDC mentioned in a figure's caption

Table 1: Quantitative segmentation results of 2D U-Net, 3D U-Net, Residual DSM and Distill DSM on BRATS 2020 dataset. ET represents Enhancing Tumor, WT represents Whole Tumor and TC represents Tumor Core

| | Class | 2D U-Net | Residual DSM | 3D U-Net | Distill DSM(Ours) |
|---|---|---|---|---|---|
| Parameters | | 1,082,211 | 1,082,211 | 4,288,208 | 1,216,266 |
| Flops per voxel | | 38,662 | 38,735 | 58,709 | 39,456 |
| Wall time per voxel(s) | | 7.9498e-7 | 8.1726e-7 | 8.6517e-7 | 8.2641e-7 |
| | ET | 0.712 | 0.732 | 0.704 | 0.753 |
| Dice | WT | 0.861 | 0.867 | 0.879 | 0.873 |
| | TC | 0.687 | 0.704 | 0.796 | 0.742 |
| | ET | 0.714 | 0.707 | 0.687 | 0.761 |
| Sensitivity | WT | 0.859 | 0.835 | 0.898 | 0.841 |
| | TC | 0.660 | 0.693 | 0.779 | 0.726 |
| | ET | 0.9997 | 0.99978 | 0.99975 | 0.99969 |
| Specificity | WT | 0.99903 | 0.99939 | 0.99896 | 0.99944 |
| | TC | 0.99975 | 0.99986 | 0.99958 | 0.9997 |
| | ET | 35.20 | 29.21 | 43.27 | 30.52 |
| Hausdorff95 | WT | 6.52 | 8.42 | 11.46 | 5.98 |
| | TC | 27.39 | 34.85 | 18.84 | 32.87 |

Figure 11: BRATS mentioned in a table's caption

**In footnotes:**

Figure 12: ACDC dataset's name mentioned in a footnote

Figure 13: CAMELYON dataset mentioned in a footnote with the URL

## Appendix D. Example of data from OpenAlex

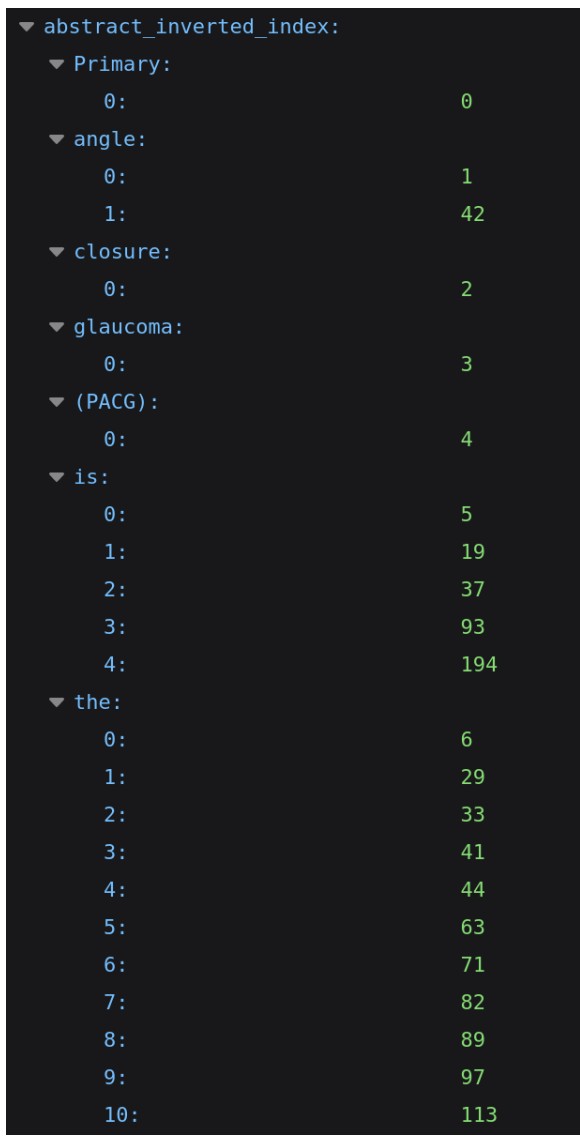

Figure 14: Example of abstract obtained from OpenAlex

```
id:                              "https://openalex.org/W3096981517"
doi:                             "https://doi.org/10.1007/978-3-030-59722-1_70"
▼ title:                         "A Macro-Micro Weakly-Supervised Framework for AS-OCT Tissue Segmentation"
▼ display_name:                  "A Macro-Micro Weakly-Supervised Framework for AS-OCT Tissue Segmentation"
  publication_year:              2020
  publication_date:              "2020-01-01"
▶ ids:                           {…}
  language:                      "en"
▶ primary_location:              {…}
  type:                          "book-chapter"
  type_crossref:                 "book-chapter"
▶ indexed_in:                    […]
▼ open_access:
    is_oa:                       true
    oa_status:                   "green"
    oa_url:                      "https://arxiv.org/pdf/2007.10007"
    any_repository_has_fulltext: true
```

Figure 15: Example of link to full text PDF obtained from OpenAlex

```
referenced_works_count:          15
▼ referenced_works:
    0:                               "https://openalex.org/W1977623353"
    1:                               "https://openalex.org/W2160605010"
    2:                               "https://openalex.org/W2165394378"
    3:                               "https://openalex.org/W2302594701"
    4:                               "https://openalex.org/W2331515946"
    5:                               "https://openalex.org/W2566969499"
    6:                               "https://openalex.org/W2613836512"
    7:                               "https://openalex.org/W2799738340"
    8:                               "https://openalex.org/W2904884925"
    9:                               "https://openalex.org/W2918552801"
    10:                              "https://openalex.org/W2963198662"
    11:                              "https://openalex.org/W2963573435"
    12:                              "https://openalex.org/W2964309882"
    13:                              "https://openalex.org/W2979907638"
    14:                              "https://openalex.org/W3102538446"
```

Figure 16: Example of list of citations in a paper obtained from OpenAlex

## Appendix E. Example of data from GROBID

Figure 17: Example of a header with abstract obtained after GROBID conversion to XML

Figure 18: Example of full text body obtained after GROBID conversion to XML

```
-<div type="references">
 -<listBibl>
  -<biblStruct xml:id="b0">
   -<analytic>
    -<title level="a" type="main">
        Pyramid Network with Online Hard Example Mining for Accurate Left Atrium Segmentation
      </title>
    -<author>
     -<persName>
        <forename type="first">Cheng</forename>
        <surname>Bian</surname>
       </persName>
      </author>
    +<author></author>
    +<author></author>
    +<author></author>
    +<author></author>
    +<author></author>
    +<author></author>
    +<author></author>
      <idno type="DOI">10.1007/978-3-030-12029-0_26</idno>
     </analytic>
   +<monogr></monogr>
    </biblStruct>
```

Figure 19: Example of a citation obtained after GROBID conversion to XML

