# OpenReview forum: "[Citation needed] Data usage and citation practices in medical imaging conferences"
_MIDL.io/2024/Conference — MIDL 2024 Oral_

### Official Review · Reviewer_E6Ls · 2024-02-22

**Confidence:** 4
**Preliminary Rating:** 4
**Recommendation:** Poster
**Final Rating:** 4

**Summary:**

The paper focuses on the usage and citation practices of datasets in medical imaging conferences. The findings demonstrate the concentration of dataset usage on a limited set of datasets and highlight different citing practices, making tracking difficult. The paper also discusses challenges in dataset identification and the impact of publicly available datasets beyond competition. The significance of the research lies in addressing the lack of an adopted standard for citing datasets, providing tools for dataset detection, and shedding light on dataset usage patterns in medical imaging research.

**Strengths:**

- This work focuses on a study of the usage of 20 publicly available medical datasets in papers from MICCAI and MIDL, providing empirical evidence on dataset usage patterns in these conferences.

- The paper also discusses challenges in tracking dataset usage, such as papers citing datasets without mentioning them and papers mentioning datasets without citing their associated papers.

- The research addresses the difficulty in tracking dataset usage and provides insights into the impact of publicly available datasets beyond competitions

**Weaknesses:**

- The study is limited to a manually selected set of datasets and venues, which may introduce bias into the findings.

- The paper acknowledges the difficulty in disambiguating between different versions of datasets, which could impact the accuracy of dataset usage tracking.

- The tools used in the study, such as OpenAlex and the PDF annotation software, may have limitations in detecting dataset usage, as they rely on specific criteria for citation and mention detection.

- The paper does not provide a comprehensive analysis of the reasons behind different citing practices, such as citing a dataset's paper without mentioning it or mentioning a dataset without citing its paper.

- The research does not explore the potential impact of dataset usage on the validity and reproducibility of medical imaging studies

**Detailed Comments:**

- The paper has to provide a comprehensive analysis of the reasons behind different citing practices, such as citing a dataset's paper without mentioning it or mentioning a dataset without citing its paper.

- Best to include the potential impact of dataset usage on the validity and reproducibility of medical imaging studies

- What are the challenges in dataset identification?

- What are the different citing practices?

- What is the impact of publicly available datasets beyond competition?

- How does disambiguation affect dataset tracking?

**Justification Of Final Rating:**

Medical imaging papers often focus on methodology, but the quality of algorithms and the validity of conclusions heavily rely on the datasets used. Creating datasets requires substantial effort, so researchers often use publicly available ones. However, there is currently no universally adopted standard for citing these datasets in scientific papers, which makes it challenging to track their usage. This study reveals a concentration in the usage of a limited set of datasets and highlights varying citing practices, complicating the automation of tracking dataset usage. The authors have addressed all of my concerns. I have no further comments.

**Justification Of The Preliminary Rating:**

The work can be valuable for researchers and reviewers in assessing the validity and reproducibility of studies. Also, the paper contributes to the understanding of dataset usage and citation practices in medical imaging, providing insights that can improve the transparency and reproducibility of research in the field.

**Questions To Address In The Rebuttal:**

1 How does one identify between citing a dataset's paper without mentioning it or mentioning a dataset without citing its paper?

2 What is the potential impact of dataset usage on the validity and reproducibility of medical imaging studies

3 What are the challenges in dataset identification?

4 What are the different citing practices?

5 What is the impact of publicly available datasets beyond competition?

6 How does disambiguation affect dataset tracking?

**Special Issue:**

No

---

> ### Author Response · Authors · 2024-03-12
>
> Thank you for your review. We addressed the comments as follows:
>
> *How does one identify between citing a dataset's paper without mentioning it or mentioning a dataset without citing its paper?* and *What are the different citing practices?*
>
> As also pointed out by another reviewer, we clarified the definitions at the beginning of Section 3. For more intuition on the different citation practices and mentions, we also added examples in Appendix C.
>
> *What are the challenges in dataset identification?* and *How does disambiguation affect dataset tracking?*
>
> The automatic identification of datasets is made difficult by a combination of large variability in names and context even for a single dataset with multiple aliases, and sometimes some close names between different datasets for example as we mentioned for COVID-19 datasets. It’s therefore complicated to know if a name is just another alias of a dataset or actually a completely different one.
>
> This problem also affects the disambiguation of datasets, a similar name between different datasets may lead to the overestimation of the usage of the one being tracked. Similarly, as different versions of a dataset may also contain different tasks or solve potential problems such as a lack of diversity in a previous version, the disambiguation could be valuable information.
>
> We extended this part of the discussion to include these remarks.
>
> *What is the potential impact of dataset usage on the validity and reproducibility of medical imaging studies?* and *What is the impact of publicly available datasets beyond competition?*
>
> The dataset can indeed impact multiple aspects, for example, the generalization of models trained and evaluated on a single dataset but with a gap in performances when applied to other datasets of the same task. Therefore, while the availability of a public dataset is a positive step towards getting a problem addressed by the community, if this dataset becomes the only point of evaluation, then performance may eventually be overestimated.
>
> We modified the introduction to add this problem.

---

> > ### Comment · Reviewer_E6Ls · 2024-03-26
> >
> > The authors have addressed all of my concerns. I have no more comments.

---

### Official Review · Reviewer_ojgU · 2024-02-27

**Confidence:** 4
**Preliminary Rating:** 3
**Recommendation:** Poster
**Final Rating:** 4

**Summary:**

The authors present  open-source tools  that could help with the detection of dataset usage; (1)  OpenAlex: a regex based full-text analysis tool, and (ii) a PDF annotation software to manually label the presence of datasets. They applied both tools to analyze MICCAI and MIDL papers in terms of citations of publicly available datasets. They aim to understand the standardization of the citations of public databases.

**Strengths:**

The subject topic is important, as public datasets play a crucial role in the rapid advances of medical data processing today.

The tools are made publicly available, and the pipeline is clearly explained.

The article is timely, as the FAIR principles in data management have become a hot topic in recent years.

**Weaknesses:**

The paper does not address the current trend of using DOIs to standardize dataset references.

It is difficult to ascertain the extent of necessary data cleanup and the level of automation achievable within the described pipeline.

The results lack clarity regarding the significance of automating the entire process.

**Detailed Comments:**

As emphasized by the authors, standardizing dataset citations is crucial for advancing medical AI through the use of public datasets. While the presented work addresses the systematic analysis of dataset citations, the effectiveness of the provided tools for achieving this goal efficiently remains unclear.

In many medical studies, similar methodologies are employed in survey studies to identify publications describing the use of specific drugs or treatments for particular diseases. This tool has the potential to be generalized for automating such efforts. The key contribution lies in its ability to automate manual and cumbersome processes. However, the authors have not clearly demonstrated the tool's effectiveness in terms of full automation or indicated the extent of necessary data cleanup.

**Justification Of Final Rating:**

All my review comments are responded by the authors. They also updated the respective section of the paper according to review comments. As all my review comments are answered, I am increasing my score in th

**Justification Of The Preliminary Rating:**

The paper addresses a significant topic in medical AI and offers valuable recommendations for best practices in dataset citations. However, the efficiency and impact of the automation tools provided by the authors remain unclear.

**Questions To Address In The Rebuttal:**

To what extent can the entire process be automated, and how much data cleanup is necessary for the results?

**Special Issue:**

No

---

> ### Author Response · Authors · 2024-03-12
>
> Thank you very much for your review.
>
> Regarding the automation of the process, it was designed to be easily used for other datasets and venues without modifying the code, therefore the main parts of the data collection and the detection of mentions/citations are automated.
>
> The following information is gathered automatically:
>
> - Get the list of paper of selected papers from DBLP
> - Get information from OpenAlex
> - Download full texts using links from OpenAlex
> - Convert PDF to XML with GROBID
> - Detect citations & mentions from OpenAlex data & GROBID
>
> The following actions need to be performed by the user (also answering your question about data cleanup):
>
> - Input the list of datasets and venues
> - Gather additional full texts for which no links were available in OpenAlex, or removing duplicate PDFs.
>
> To give an example, if users want to perform the same analysis but including the papers from MIDL 2024, they would have to change the maximum year in venues.csv and could use the same scraping tool without code modification but by adding the volume number of MIDL 2024 in the command line to launch it.
>
> Overall, no code modification is needed for the data collection and processing except for an optional scraping tool that users could define to fit their needs. Note that OpenAlex may be sufficient to obtain the full text and therefore such a scraping too may not be necessary.
>
> We added details about the data cleanup in section 3.

---

> > ### Comment · Reviewer_ojgU · 2024-03-26
> >
> > Thanks for the responses to the review comments. As the authors added the details to the submission I am increasing my score.

---

### Official Review · Reviewer_M2fw · 2024-02-28

**Confidence:** 3
**Preliminary Rating:** 3
**Final Rating:** 4

**Summary:**

In this meta-research, the authors investigate the adequacy of data citation and mention practices in medical imaging research papers, proposing a novel pipeline and open-annotation tool for this analysis. Focusing on publications from the MICCAI and MIDL conferences between 2013-2023, they provide an evaluation of citation behaviors. The findings highlight significant insights into the current state and potential areas for improvement in research integrity and data usage acknowledgment within the field.

**Strengths:**

Large corpus of papers examined from two major conferences.

Great background section, helpful and informative.

Findings elucidate important issues potentially contributing to the reproducibility crisis.

**Weaknesses:**

Although the terminology starts off seeming straightforward, clarity becomes an issue as the paper progresses, leading to confusion.

The explanation of external tools used in the study is not sufficient, making the full pipeline difficult to understand.

**Detailed Comments:**

I would be really interested to see these trends in brain research.
By-product findings are interesting and helpful.

Minor comments:
1. In the Introduction section, the sentence that starts with, "There is a need to analyse research within a field ..."   this sentence could be rephrased.
2. Discussion section under anecdotal findings: Such page restrictions may incentivize ... > Relaxing such page restrictions ...
3. In the methods development, it is common to use the same datasets (not only) so the baseline comparison is achievable. This potentially contributes to popular datasets being used a lot more frequently and consequently inflating the numbers. Excluding methodology papers, challenges and focusing on application studies potentially could help with the inflated numbers.

**Justification Of Final Rating:**

The authors have addressed each of my comments and provided discussions. I've decided to increase my initial score, believing it accurately reflects the paper's quality and merits within the context of its field.

**Justification Of The Preliminary Rating:**

Interesting research question, relevant and useful for reproducibility of our research. However some work needs to be done to improve the clarity of the paper, if the authors work on the minor issues suggested under the "Questions To Address In The Rebuttal", I think the paper should be in the conference and is of interest to the community.

**Questions To Address In The Rebuttal:**

It would be helpful to the reader if the authors:
- explicitly define key terminologies such as "usage," "mention," and "citation," outlining their distinct meanings within the context of this paper.
- Section 5.1 Data Selection par. #4 "All the papers ...", clarify how the modular part (I think) in the additional scraping tool help MIDL paper and why this was not used for MICCAI papers.
- Figure 2, if possible add "actual usage" to the figures.
- explain briefly in the methods section how OpenAlex and GROBID work.

What are some of your insights on:
- data derivatives (if the data used is not coming from the original data source) and how they should be documented/cited?
- "dataset accessibility" as a key factor?

---

> ### Author Response · Authors · 2024-03-12
>
> Thank you very much for your review.
>
> *It would be helpful to the reader if the authors:*
>
> - *explicitly define key terminologies such as "usage," "mention," and
> "citation," outlining their distinct meanings within the context of this paper.*
> - *Section 5.1 Data Selection par. #4 "All the papers ...", clarify how the modular part (I think) in the additional scraping tool help MIDL paper and why this was not used for MICCAI papers.*
> - *Figure 2, if possible add "actual usage" to the figures.*
> - *explain briefly in the methods section how OpenAlex and GROBID work.*
>
> To improve clarity we did the following:
>
> We rearranged Section 3 to include the definitions at the beginning and we also added some examples of citations and mentions in Appendix C.
>
> We also added more explanation on the modular part and why we did it for MIDL and not MICCAI in section 5.1
>
> Unfortunately, the number of papers being too high (2521 papers) we did not screen all of them it’s therefore not possible to add the actual usage in the figure, however while developing the method we saw that the heuristics we use (for example a dataset mentioned in the methods section) often indicates actual use.
>
> We added a small description of OpenAlex (aggregating information from other sources) and GROBID (a cascade of models to extract information) in Section 3. We also added an example of data from both tools in Appendix D and E in case it helps to understand the process.
>
> We hope the changes make the paper and the process easier to understand.
>
> *What are some of your insights on:*
>
> - *data derivatives (if the data used is not coming from the original data source) and how they should be documented/cited?*
>
> This is a great question, it is difficult to say something general about this because there are many different degrees of derivatives. For example, the LUNA16 challenge uses selected cases from the LIDC-IDRI dataset, however, papers may still talk about the “LUNA16 dataset” and only refer to the challenge website. Since the challenge website clearly documents that the data is based on LIDC-IDRI, and the data license allows it, only describing the actual data used and then citing LUNA16 seems sufficient.
>
> In other cases it is less clear. Suppose the experiments use two independently released derivatives B and C, of some original dataset A. Then it would be important to document that both derivatives were based on A (and therefore to cite it), since there might be overlaps in B and C. But at some point, if multiple original datasets were involved, this might become impractical.
>
> Due to space limitations it is difficult to go in-depth into this point at the paper but it is clearly an important question that we should discuss as a community, we added a note about this to the Discussion section.
>
> - *"dataset accessibility" as a key factor*
>
> One of our original questions was to study what factors lead to increased dataset use, and likely availability/accessibility is strongly linked to this. Downloading a (copy of) a dataset from Kaggle is simpler than filling in a data agreement. But while this increases accessibility/use, we feel it might decrease our capacity to build knowledge, since in this process many details about the data can be lost - see for example another recent work from our lab https://arxiv.org/pdf/2402.06353.pdf, we added a note about this in the Discussion.

---

> > ### Comment · Reviewer_M2fw · 2024-03-26
> > **Response to Rebuttal**
> >
> > Thank you for your response addressing my concerns and discussions on derivatives.

---

### Official Review · Reviewer_dAYf · 2024-02-29

**Confidence:** 4
**Preliminary Rating:** 4
**Recommendation:** Poster
**Final Rating:** 4

**Summary:**

The authors studied citation and mention patterns of medical AI papers and commonly used datasets. The authors analyzed both citations and text of publications to identify datasets that are being used or mentioned.

**Strengths:**

The authors present research in an unique area around tracking and summarizing use of datasets in medical AI research.

The authors demonstrated feasibility of its pipeline on 20 commonly used medical datasets.

Authors also studied trends and most commonly mentioned datasets that could have implication for the field.

**Weaknesses:**

Whether mentioning a dataset vs using the dataset for train/validation/test could also be an interesting topic that is underexplored in the project.
What would be interesting is also including a way to track emergence of new dataset and their impact on healthcare AI.

**Detailed Comments:**

Overall I think this is an interesting straghtforward project touches on an important consideration for our field.

**Justification Of Final Rating:**

I appreciate the authors' update and explanations to my questions and suggestions. Overall I found the project very interesting and an important research area. I think additional implications of the research results and actionable next steps are important for authors to consider and continue the work on.

**Justification Of The Preliminary Rating:**

Overall I think this is an elegant paper that tracks an important aspect of our research world and would be helpful to be a bit larger in scope and a bit more rigorous in methods. Overall I recommend acceptance of this work as a poster.

**Questions To Address In The Rebuttal:**

Evaluation of OpenAlex's accuracy and completeness in this setting? I may have missed this but seem an important prerequisite.
Evaluation of correlation of multiple datasets used in the same publicatoin?

**Special Issue:**

No

---

> ### Author Response · Authors · 2024-03-12
>
> Thank you very much for your questions. To start with the second one about the correlation of multiple datasets, we indeed computed it with a co-occurrence matrix. We found very low numbers of papers with such co-occurrences, and both due to the low sample size and space limitations, we decided to not include this result in the paper.
>
> Regarding the evaluation of OpenAlex, it’s indeed an important prerequisite, especially for the detection of citations. However, we mitigate this dependence with the citation detection in our tool using GROBID and it could be more developed to reduce this risk.
>
> We also compared OpenAlex with another open citation index tool named OpenCitations. We compared the citations returned for our set of datasets on cardiac segmentation. From this comparison, 97% of the citations returned by OpenCitations were also returned by OpenAlex. On the other hand, only 84% of the citations returned by OpenAlex were returned by OpenCitations.
>
> We updated the paper to include the comparison with OpenCitation at the end of the first paragraph of section 3.

---

> > ### Comment · Reviewer_dAYf · 2024-03-26
> > **Thank you**
> >
> > Thank you for your thoughtful responses to my questions and comments. I do not have further questions.

---

### Author Response · Authors · 2024-03-12

Thank you to all the reviewers. We uploaded a revised version of the paper and we will shortly answer your individual questions. To summarize, the main changes are:

- We clarified the different terms (citation, mention and usage) in Section 3. We also added some examples of citations and mentions in the Appendix.
- We added a brief description of OpenAlex and GROBID and explained why we chose OpenAlex. We also included an example of data from OpenAlex and GROBID in the appendix.
- We added some notes in the discussion about derivative datasets and the impact of disambiguation.
- We removed Table 1 with the number of papers in different groups, and described this within the text, to comply with the page limit.
- Other minor text changes to comply with the page limit.

---

### Author Response · Authors · 2024-03-22

Dear area chair and reviewers,

Thanks again for your input on the paper. It's been 10 days since we revised the manuscript and answered the reviewers' questions. With the discussion period ending next week, we were wondering if there are any responses to our changes, is there anything else we need to clarify?

---

> ### Comment · Area_Chair_PfdP · 2024-03-22
>
> Dear authors,
>
> Thanks for reaching out, and for posting your rebuttals with ample time. Please note that this year, the "rebuttal" and "discussion" phase are separated, so the discussion phase, where reviewer can interact with authors, has only started four days ago, and will last until 27 March.
>
> However, I do encourage the reviewers to respond to the author comments as soon as possible to enable a constructive discussion.
>
> All the best,
> AC

---

### Comment · Area_Chair_PfdP · 2024-03-26
**Concerns addressed?**

Dear reviewers,

Could you please briefly comment whether your concerns were adequately addressed by the rebuttal, and if necessary or helpful, engage in a discussion with the reviewer?

Thank you very much, and best wishes,
AC

---

### Meta-Review · Area_Chair_PfdP · 2024-04-02

**Recommendation:** Accept (Poster)
**Confidence:** 5

**Metareview:**

The reviewers believe (and I agree) that this paper addresses an important topic for our community. The authors have done a good job addressing the initial reviewer comments, and after the revisions all reviewers have reached a consensus and recommend acceptance of the paper.

---

### Decision · Program_Chairs · 2024-04-06

Accept (Oral)